# The Detection of SARS-CoV-2 in the Environment: Lessons from Wastewater

Tasha Marie Santiago-Rodriguez

Diversigen, Inc., Houston, TX 77046, USA; trodriguez@diversigen.com

**Abstract:** Wastewater has historically been an important source of enteric pathogens, as well as a source of unconventional or unexpected pathogens, including those present in the respiratory tract, saliva, urine, and blood. This is the case with SARS-CoV-2, the causative agent of the most recent pandemic. SARS-CoV-2 has been identified in wastewater across various geographical regions prior to, and during, the report of cases. The detection of SARS-CoV-2 in wastewater is usually performed using molecular techniques targeting specific genomic regions. High-throughput sequencing techniques, both untargeted and targeted or amplicon-based, are also being applied in combination with molecular techniques for the detection of SARS-CoV-2 variants to determine the genetic diversity and phylogenetic relatedness. The identification of SARS-CoV-2 in wastewater has a number of epidemiological, biological, and ecological applications, which can be incorporated into future outbreaks, epidemics, or pandemics.

**Keywords:** amplicon sequencing; high-throughput sequencing; SARS-CoV-2

## 1. Environmental Surveillance of Pathogens from Wastewater

Wastewater is a source of enteric pathogenic microorganisms that can represent a concern to public health [1]. Enteric pathogenic bacteria, viruses, protozoa, and parasites can be present in wastewater, and these pose the risk of being transmitted through water sources (i.e., water-borne), particularly in cases where wastewater effluents have not been efficiently treated before their discharge into water streams that are used for recreation and consumption [2,3]. Untreated wastewater usually includes fecal waste from built environments, as well as waste from other sources, including, but not limited to, rainwater and waste from industrial use. Zoonotic pathogens may also be transmitted through wastewater [4], some of which can cause life-threating conditions, particularly in developing countries with no immediate access to health care. For these reasons, wastewater has been historically used for the surveillance of enteric pathogens.

The surveillance of enteric pathogens usually occurs in clinical settings, which limits results to symptomatic individuals seeking treatment and testing. This approach may also limit surveillance efforts by ignoring asymptomatic individuals, who may also be shedding pathogens. For this reason, environmental surveillance using wastewater has been proposed and is considered as an approach to identify potential early signs of outbreaks, enabling decision-making in a timely manner. While the environmental surveillance of enteric pathogens in wastewater is not entirely novel, it is still underestimated and underused, partly because more research is needed to understand its sensitivity. Environmental surveillance has been used to detect several pathogens. For instance, wastewater surveillance has been performed during poliovirus outbreaks, where results have yielded a dose-dependent relationship between the number of poliovirus shedders and the amount of poliovirus in wastewater [5]. This approach illustrates the application of the environmental surveillance of pathogens, which could further open the opportunity to evaluate containment efforts, as needed. Another example includes the detection of the hepatitis A and noroviruses

outbreaks. While these viruses may be excreted in high concentrations in infected individuals, the presence of these viruses in wastewater may be the early warning signs of an outbreak [6].

## 2. Detection of Enteric Pathogens in Wastewater

Historically, the environmental surveillance of enteric pathogens has been performed using culture methods. Specific media could often be used for the detection of the pathogen(s) of concern; however, culturing pathogens may be time-consuming, as it may take days to obtain results in certain cases (e.g., viruses). In addition, given that the number of some enteric pathogens is usually low, methods for the recovery, concentration, and enumeration can be troublesome [7]. For instance, cell cultures have been used for the detection of viruses in wastewater. However, it is virtually impossible for cell cultures to support the replication of all virus strains and variants. In addition, factors such as the passage number and the sequential passage of a sample in cell culture can affect results, and may underestimate virus infectivity [8]. For these reasons, molecular methods have been developed for the timely and specific detection of the pathogen(s) of concern, facilitating the identification of potential sources of contamination, and further preventing risks to public health.

Molecular methods, particularly the polymerase chain reaction (PCR) (which can determine presence/absence of a target genomic region) and quantitative PCR (qPCR) (which can quantify the copies of a target genomic region) have long been used for the detection of pathogens in wastewater [9]. Unlike culture methods, the detection of a pathogen in wastewater using PCR and PCR-based methods can provide results in hours; however, one drawback of many PCR and PCR-based methods is that they may not necessarily address infectivity [8], which may cause an overestimation of the targeted pathogen(s). Specificity may also be affected by the reagents used during PCR or qPCR amplification. For instance, polymerases with varying levels of error-proof amplifications can affect results. Taq polymerases, for example, are ideal for routine PCR, but they may fail with the amplification of genomic targets larger than 1.5 kilobases (kb) [10]. In addition, while other polymerases are ideal for cloning, others may be better suited for mutation identification, a factor that is essential in experiments aiming to distinguish pathogenic variants and strains [10].

## 3. Unexpected Viruses in Wastewater

While the gut of warm-blooded animals is inhabited by bacteria, protozoa, parasites, and viruses (which can infect bacteria, archaea, and small eukaryotes, as well as human and animal cells), human and animal viruses that can cause disease are amongst the most concerning. Enteric pathogenic viruses are easily transferred from person-to-person and require very low infectious doses to cause disease [11]. For instance, 10–100 rotaviruses are needed to cause disease [12], and $10^{12}$ virus particles per gram of stool may be shed from infected individuals [13]. Viral pathogens transmitted through the fecal-oral route can often be the focus of wastewater research, as they replicate in the gut of warm-blooded animals and are shed through the feces of infected individuals, often in high concentrations. Many of these include, but are not limited to, rotaviruses, adenoviruses, polioviruses, enteroviruses, and noroviruses [13].

Interestingly, viral pathogens that can be transmitted through person-to-person contact, aerosol droplets, and contaminated fomites [14], have been found to survive in wastewater [15]. For instance, studies addressing the prevalence of Ebola in wastewater have shown that the virus can be detected for at least the duration of the study (8 days) [16]. Similarly, while the severe acute respiratory syndrome coronavirus (SARS-CoV) is not widely distributed, and is highly prevalent in sewage [17], it has been detected in the feces of infected individuals [18]. While some of the mentioned viruses reside in the respiratory tract, it is usually presumed that they may be present in low concentrations in the gastrointestinal tract and, therefore, shed into water systems. Some of these viruses can also be

shed through saliva, phlegm, urine, and blood [19], which would also enter the wastewater system. Thus, it would be feasible for wastewater and drinking water plants to increase their scrutiny to prevent potential means of transmission.

## 4. Emerging Respiratory Viruses in Wastewater: Considerations from SARS-CoV-2

SARS-CoV-2 is the most recent virus from the *Coronaviridae* family known to infect humans, and it started as a pneumonia of unknown etiology [20]. SARS-CoV-2 is hypothesized to be of zoonotic origin, presumably originating in bats, with pangolins as intermediate hosts before adapting and infecting humans [21]. When infected, individuals may present symptoms that include a fever, cough, difficulty breathing, as well as other less-common symptoms, including vomiting and diarrhea [20,22]. SARS-CoV-2 nucleic acids (i.e., RNA) have been detected in the feces of infected individuals [23]. In addition, it is known that masks used during the pandemic may also reach water systems, many of which can harbor detectable levels of SARS-CoV-2 RNA [24]. While there is no evidence, thus far, suggesting that direct contact with feces and sewage harboring SARS-CoV-2 may cause the disease, future studies regarding potential transmission should not be ignored [25]. Nevertheless, the presence of SARS-CoV-2 in feces prompted the investigation of the prevalence of the virus in wastewater (Figure 1) [26–28].

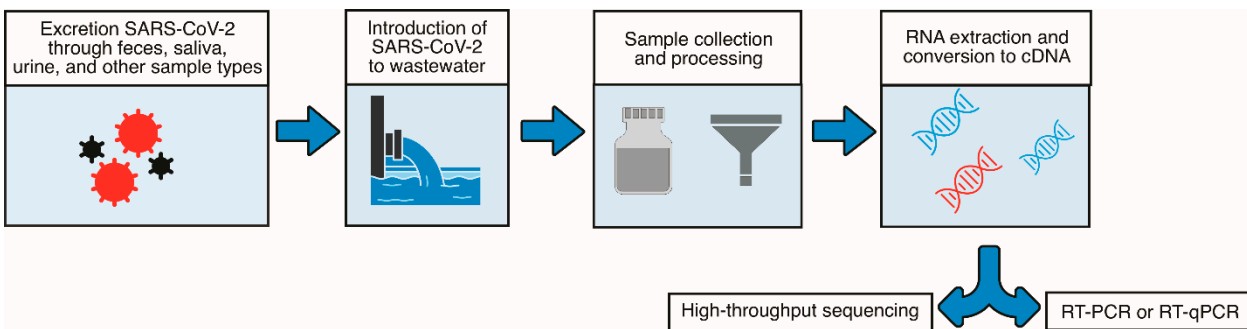

**Figure 1.** Overall process for the detection of SARS-CoV-2 in wastewater. SARS-CoV-2 may be present in the feces of infected individuals. SARS-CoV-2 may also be in other sample types, including phlegm and saliva, as well as other materials, such as masks, all of which reach the wastewater system. Samples may be collected for the detection of the virus, and samples may be pre-processed prior to RNA extraction and analysis. Methods for the detection of SARS-CoV-2 in wastewater usually involve reverse transcription PCR (RT-PCR). In addition, non-conventional methods, such as high-throughput sequencing, have been applied, offering other advantages that RT-PCR alone may not provide.

Similar to the detection of enteric viruses in wastewater, the identification of SARS-CoV-2 in such a sample type comes with several advantages. For instance, wastewater surveillance of SARS-CoV-2 may represent a cost-effective method to understand the onset of its infection and transmission within specific communities. SARS-CoV-2 wastewater surveillance may provide real-time epidemiological data that may not be provided by healthcare facilities alone [29]. Access to health care may be limited for certain communities and this may restrict the number of reported SARS-CoV2 cases, which may be limited to symptomatic individuals seeking a diagnosis. In addition, SARS-CoV-2 wastewater surveillance has the potential to capture both symptomatic and asymptomatic infections within a community before being reported. Notably, it has been suggested that SARS-CoV-2 wastewater surveillance may lead diagnostic tests by approximately a week [29]. The reason for this is that the shedding of SARS-CoV-2 may occur immediately after infection. SARS-CoV-2 monitoring in wastewater also possesses the potential to identify new variants in real-time. Indeed, while new SARS-CoV-2 variants are usually detected in clinical specimens, wastewater surveillance has been an indication of the presence of specific variants when very low cases have been reported [30]. This may have further repercussions in terms of

hospital and emergency room visits if SARS-CoV-2 wastewater surveillance is evaluated regularly [31]. In addition, the identification of SARS-CoV-2 in wastewater, using specific methods (i.e., sequencing, as discussed below), has the potential to track virus evolution. Although studies usually identify new variants in clinical specimens, wastewater may represent a source for tracking SARS-CoV-2 evolution prior to, or at the beginning stages of, the identification of new variants in clinical specimens.

The first study on the presence of SARS-CoV-2 in sewage was conducted in The Netherlands, before the onset of the epidemic in the country [27]. Wastewater treatment plants were selected that served two large cities, three medium-sized cities, and an airport. Samples were collected starting three weeks before the first SARS-CoV-2 case was identified in The Netherlands [27]. RT-qPCR results showed that the SARS-CoV-2 copy numbers increased in wastewater samples as more cases were identified [27]. In the United States, 2 out of 15 (13%) untreated wastewater samples tested positive for SARS-CoV-2 nucleic acids when using RT-qPCR. Notably, the secondary-treated wastewater and the final effluent samples tested negative for SARS-CoV-2 nucleic acids, showing that SARS-CoV-2 is sensitive to the wastewater treatment process [32]. Similarly, the wastewater samples in Italy, one of the countries to be the most affected at the beginning of the pandemic, were also positive for SARS-CoV-2 nucleic acids. Results showed that 6 out of the 12 untreated wastewater samples collected between February and April 2020 were positive for the virus nucleic acids [33]. Past and ongoing studies have shown similar results with the various SARS-CoV2 variants in countries including, but not limited to, Spain [34], Australia [28], Canada [35], and Mexico [36]. More recently, several unpublished studies have identified SARS-CoV-2 B.1.617.2 (Delta) and B.1.1.529 (Omicron) variants in wastewater before the identification of symptomatic and asymptomatic cases. While the results may suggest that SARS-CoV-2 in untreated wastewater may serve as an early warning sign [28], its absence in treated wastewater may suggest that the transmission of the virus via this source may be limited. Further studies are also needed to understand the prevalence of SARS-CoV-2 in wastewater in developing countries to understand the potential hotspots and wastewater treatment efficiencies. More studies are also needed to understand the survival and inactivation rates of SARS-CoV-2 in wastewater, which could help elucidate potential transmission mechanisms.

## 5. Wastewater Sampling Strategies for SARS-CoV-2 Detection

The wastewater surveillance of SARS-CoV-2 should be based on public health needs. Surveillance efforts should be coordinated by research laboratories, as well as public health authorities, to ensure that sampling strategies are driven by public health needs and that the results are integrated with other sources of surveillance information. Data from the surveillance of SARS-CoV-2 in wastewater could be used as a potential early warning sign of infection within a community. Both symptomatic and asymptomatic individuals might be reservoirs of SARS-CoV-2 and may potentially shed the virus through feces and other sample types that may, consequently, reach the wastewater system. Notably, the detection and quantification of SARS-CoV-2 RNA in wastewater is not an indication of the number of viable viral particles or the number of individuals that might be infected [28].

As described above, data have shown that SARS-CoV-2 is usually not present in treated wastewater. For this reason, untreated wastewater and sludge are currently the sample types being considered for surveillance studies (Figure 1). Untreated wastewater may be sampled from a wastewater treatment plant's influent, or upstream in the wastewater collection network. Moreover, it has been recommended to collect samples when defecation happens the most frequently, which is usually early in the morning [37]. There are several collection methods, which include grab and composite samples. While grab samples do not require specialized equipment, they only represent a single time point, which may not be representative of the global fecal composition of sample sites [38]. Composite samples, on the other hand, represent multiple grab samples and may improve representation efforts. Selecting between the grab or composite samples will depend, therefore, on the aims of the

study and the length of the surveillance efforts. In terms of the collection frequency, it has been suggested to collect > 2 composite samples per week, or a composite sample every 48 to 72 h, as SARS-CoV-2 is not consistently shed from infected individuals [37].

## 6. Wastewater Transport and Storage for SARS-CoV-2 Detection

Few wastewater surveillance studies have reported the transport and storage conditions of samples prior to SARS-CoV-2 detection [37]. This is particularly problematic when attempting to perform comparative analyses on the prevalence and persistence of the virus across various geographical sites. Ideally, wastewater samples for SARS-CoV-2 detection should be transported following the requirements for the identification of any enteric indicator of fecal contamination or pathogens. Briefly, samples should be collected in appropriately sterile containers, either factory-sealed or reused, and usually at least one liter of the sample should be collected [39]. Quality assurance usually may involve field blanks, which may include sterile water in sterilized containers, which are later processed along with the field sample; internal duplicates (i.e., a duplicate sample collected at the same time and place by the sampler or by another sampler); and external duplicates (i.e., a duplicate sample collected and processed by an independent sampler or team at the same time and place). It is strongly recommended to keep samples on ice or at 4 °C, and concentrated within a time frame of 48 to 72 h post-collection [39,40]. However, the inactivation rate data of SARS-CoV-2 have shown that the virus can persist for approximately 8 to 28 days in untreated wastewater. This may suggest that, under certain circumstances, samples may be processed up to 72 h post-collection with the caveat that viral RNA may not reflect the original viral load [41].

## 7. Concentration of Wastewater Samples for SARS-CoV-2 Detection

Pathogenic viruses in wastewater may be present in lower concentrations compared to fecal specimens; therefore, wastewater samples may need to be concentrated prior to nucleic acid extraction and virus detection. Studies assessing SARS-CoV-2 detection in wastewater, specifically, have utilized one, or a combination of, various concentration methods. Several of the initial studies detecting SARS-CoV-2 RNA utilized a combination of centrifugation to remove big debris, followed by the ultrafiltration of the supernatant [42]. Other subsequent studies have tested various methods, including adsorption–elution using an electronegative membrane [32,43]; adsorption–extraction with an acidic and neutral pH, as well as 25 mM $MgCl_2$; polyethylene glycol (PEG 8000) precipitation; ultracentrifugation, followed by protein column concentration; and serial centrifugation rounds [44]. Comparative studies have shown that the most efficient methods for SARS-CoV-2 concentration are the adsorption–extraction method with $MgCl_2$ and the adsorption–extraction method with a neutral pH [44]. Such comparative studies assessing the efficiency of concentration methods for SARS-CoV-2 and any emerging viruses will continue to be essential and tested as needed.

## 8. Nucleic Acid Extraction and Purification from Wastewater Samples for SARS-CoV-2 Detection

Several methods exist for the extraction of SARS-CoV-2 RNA from wastewater once the sample has been concentrated. While commercially available methods may be more consistent and standardized for the extraction of RNA from SARS-CoV-2, the pandemic has resulted in a shortage of extraction kits. Alternative methods, consisting of reagents found in any molecular diagnostic laboratory, have been developed and tested and are comparable to commercially available RNA extraction kits. Several of these methods include BSA, TRIzol, and acid pH treatment, and have produced high yield of SARS-CoV-2 RNA [45]. Several treatments, however, may not be as straightforward to implement as they require a chemical hood as part of the extraction; thus, each laboratory would have to evaluate the most suitable RNA extraction method(s) depending on resource availability [45]. Prior any further analyses, it is also essential to ensure that high-quality RNA has been obtained, so

biases are not introduced due to low-quality RNA. Cross-contamination can occur when RNA extraction is performed, either manually or in an automated fashion. For this reason, a reagent blank, also known as a negative extraction control, should be included for each batch of RNA extractions to ensure no cross-contamination has occurred [37].

## 9. Molecular Methods for SARS-CoV-2 Detection in Wastewater: The PCR-Based Method

PCR-based methods are preferred for the rapid identification of SARS-CoV-2 in wastewater samples. The most common PCR method for the detection of SARS-CoV-2 in wastewater, and in other sample types, is reverse transcription PCR (RT-PCR) and quantitative RT-PCR (RT-qPCR) (Figure 1). RT-PCR and RT-qPCR provide several advantages for the detection of SARS-Co-2 in wastewater, as well as other sample types, including, but not limited to, a wide availability, the viral load quantification, and the early and rapid detection of the virus. RT-PCR and RT-qPCR are employed when the RNA is the starting material, and it is followed by the transcription of RNA into its complementary DNA (cDNA). The cDNA is then used as the template for the qPCR reaction.

As with any PCR-based method, its success will depend on the reagents used. For instance, primer specificity is essential when performing RT-PCR and RT-qPCR; therefore, the RNA target genome(s) should be known in order to increase sensitivity [46]. The reaction can also be affected by inhibitors, which can be concentrated when concentrating the sample. One way to address PCR inhibition is by spiking known viral particles of a similar morphology and genetic composition to the virus of interest at a specific concentration. PCR inhibition should be assessed before making assumptions regarding the presence/absence and/or viral RNA copy numbers [37]. Another important factor that could affect the detection of SARS-CoV-2 RNA in any sample type, including wastewater, is the enzymes. The gold standard for a SARS-CoV-2 diagnosis is the TaqMan-based RT-PCR and RT-qPCR. However, TaqMan-based assays may be cost-prohibiting to many laboratories around the world who have a limited access to PCR reagents. For this reason, studies have evaluated the use of SYBR-Green assays as an alternative, cost-efficient method for the detection and diagnosis of SARS-CoV-2 [47]. Results have shown positive reactions with all serial dilutions of the SARS-CoV-2 isolate tested [47]. However, SYBR-Green assays have not widely been evaluated for the detection of SARS-CoV-2 nucleic acids in wastewater.

In addition, when testing for the presence SARS-CoV-2 nucleic acids in any sample type, it is of utmost importance to keep the proper quality control standards when using RT-PCR and RT-qPCR. Since the assays may mostly be performed in multi-well plates, there is a risk of aerosol cross-contamination. For this reason, a no-template control, which may include nuclease-free water and a proper negative, should be included in the amplification assay(s). Positive controls should also be included, which should provide a recommended Ct value (threshold cycle), as recommended by the manufacturer's instructions [48].

## 10. Molecular Methods for SARS-CoV-2 Detection in Wastewater: High-Throughput Sequencing

High-throughput sequencing has been used to characterize the viral composition of wastewater samples [49]. While the origin and further adaptation of SARS-CoV-2 remains a matter of further research and speculation [21], high-throughput sequencing, specifically RNA-sequencing or meta-transcriptomics, was originally used in combination with other molecular techniques to identify the novel coronavirus [20]. High-throughput sequencing is a technique used to identify the genome sequence(s) of organisms by massive sequencing. After sequencing, results are usually compared to a reference database containing genomes of interest, and the comparisons can be made using various approaches. One of the approaches involves looking at the similarity across the genomes. For the discovery of the novel coronavirus, specifically, lower respiratory tract samples were collected from patients with a pneumonia of an unknown etiology. A combination of high-throughput sequencing and a RT-PCR assay, targeting a consensus region of a specific group of coronaviruses, known as beta coronaviruses, was used to discover the unknown virus. Thousands of

sequences were approximately 85% similar to the coronavirus found in bats, demonstrating differences from the previous SARS-CoV and MERS-CoV sequences [20]. This study demonstrated the feasibility of high-throughput sequencing to continue to identify novel coronaviruses and expand the comparison capabilities. High-throughput sequencing also arose and has been proposed as a surveillance tool for the detection of SARS-CoV-2 in various sample types, including wastewater, and as a means to determine genetic diversity (Figure 1) [50–52].

*Untargeted vs. Targeted (Amplicon-Based) High-Throughput Sequencing*

Generally, there are two types of high-throughput sequencing approaches for the detection of SARS-CoV-2 in wastewater and in other sample types: untargeted and targeted, or amplicon-based (Figure 2). Untargeted high-throughput sequencing refers to the sequencing of a sample without the intention of targeting any organism, whereas targeted or amplicon high-throughput sequencing usually relies on primers that target specific strains and variants. When applying these sequencing approaches for the detection of SARS-CoV-2 in wastewater and in other sample types, both sequencing approaches require RNA extraction after the samples have been collected and pre-processed as needed, and RNA is then converted to cDNA. In the case of targeted or amplicon high-throughput sequencing, cDNA is amplified using multiplex PCR with overlapping primers, which amplify most of the viral genome. Multiplex PCR with overlapping primers is not applied in untargeted high-throughput sequencing. In both sequencing approaches, cDNA then proceeds to library preparation and sequencing (Figure 2). Once there, the targeted or amplicon sequencing sequences need to be 'stitched' or merged. This is not necessarily the case for untargeted high-throughput sequencing, as the read assembly may not be required for database interrogation and further characterization.

Untargeted high-throughput sequencing is usually employed for virus discovery and for determining genetic diversity; however, untargeted high-throughput sequencing could also be employed for virus surveillance if the database used for the analysis contains the reference genome of interest, or a phylogenetically close variant or strain (Figure 2). As mentioned above, untargeted high-throughput sequencing, in combination with other techniques, has been used for the discovery of SARS-CoV-2 and has been continued to be used to obtain SARS-CoV-2 genomes from various origins. However, most of the SARS-CoV-2 surveillance in wastewater employs assays targeting known SARS-CoV-2 variants, often showing that these are identical to their clinical counterparts [51]. The high-throughput sequencing of wastewater samples looking for SARS-CoV-2 sequences may provide a degree of genetic diversity [53]. The amount of data provided in untargeted high-throughput sequencing may also enable the study of the wastewater microbiome, virome (the group of all viruses), and, potentially, the resistome (the group of antibiotic-resistance genes) in association with SARS-CoV-2. However, the high-throughput sequencing of wastewater samples may have the caveat of not providing the same level of genome confidence as the sequencing of clinical samples, as these are highly diverse samples, and more sequencing may be needed to obtain a higher degree of sequence coverage [52].

The targeted or amplicon-based high-throughput sequencing for SARS-CoV-2 detection has been mostly employed in clinical samples. The method involves the amplification of the SARS-CoV-2 genome in segments, which are then sequenced, 'stitched' together by finding overlapping regions, and are compared to existing SARS-CoV-2 variants (Figure 2). Companies and institutions have developed unique and specific approaches to circumvent untargeted high-throughput sequencing, mostly because it reduces the amount of data that need to be analyzed. Unlike untargeted high-throughput sequencing, targeted or amplicon-based approaches only require < 1 million reads to gain insights into the SARS-CoV-2 prevalence and variant genomic information [54]. The potential caveat of amplicon-based approaches for SARS-CoV-2 detection is that the genomic ends may not be covered; thus, 100% genome coverage may not be reached. However, genome recovery can usually be attained at around > 99.0%, which may be sufficient for phylogenetic

relatedness analyses [55]. While more studies are still needed to assess the effectiveness of targeted high-throughput sequencing approaches that target SARS-CoV-2, studies have evaluated this technique in wastewater samples. For instance, amplicon-based sequencing data, obtained from a total of 48 wastewater samples collected from wastewater treatment plants in Switzerland between 8 July and 21 December 2020, showed specific lineages and circulations within the communities explored [56]. More recently, the amplicon-based sequencing of SARS-CoV-2 variants has been performed in various geographical regions. The results from this study revealed over 100 mutations that are categorized into 39 types of mutations [57]. The amplicon-based sequencing of SARS-CoV-2 is usually based on the ARTIC protocol, found in https://artic.network/ (accessed on 24 January 2022). Modifications to the ARTIC protocol continue to be tested in sample types, such as saliva; thus, their efficiency in wastewater would need to be tested [55,58].

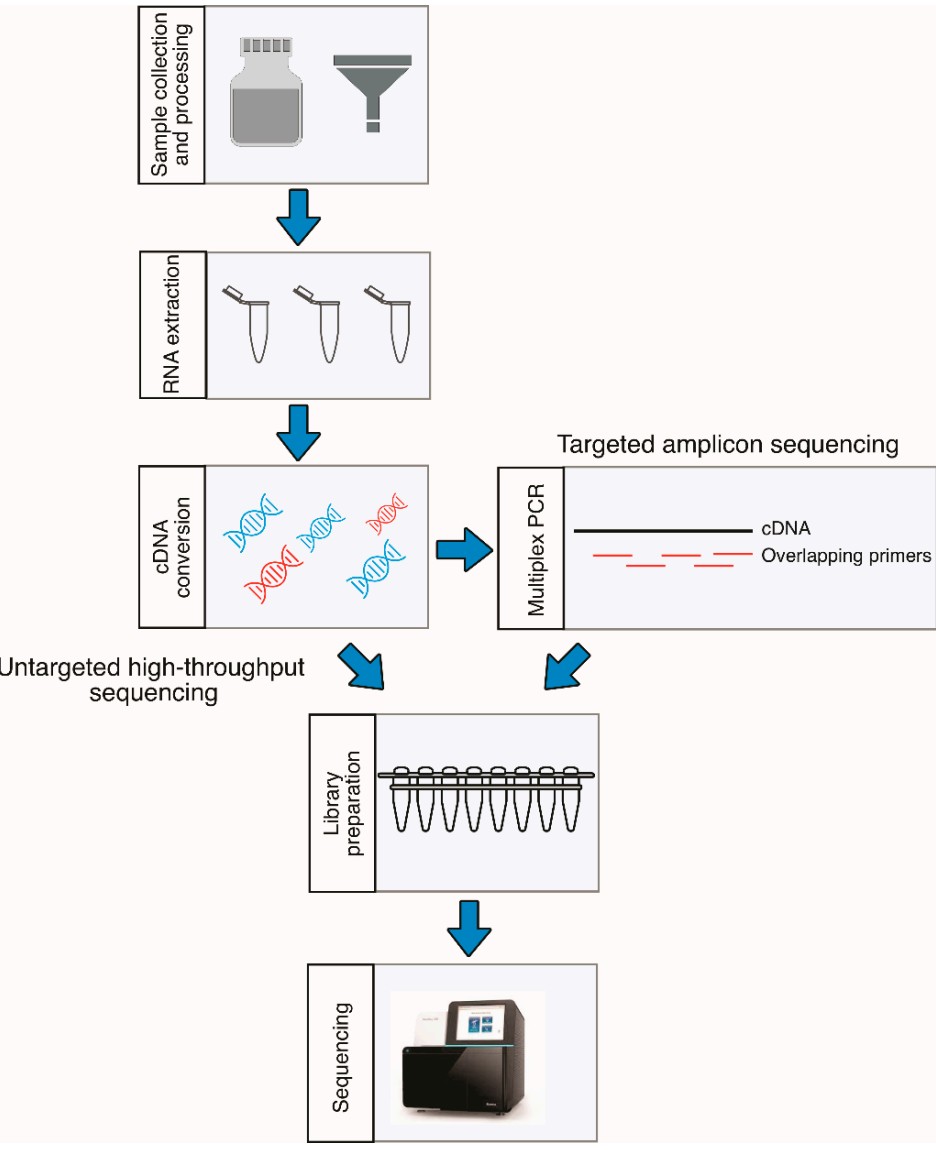

**Figure 2.** General steps for sample processing for untargeted and targeted or amplicon high-throughput sequencing. After sample collection and pre-processing (see text), RNA may be extracted and converted to cDNA. For targeted amplicon sequencing, cDNA is amplified using multiplex PCR with overlapping primers that target most of the viral genome. Product is then processed for library preparation and sequencing. Alternatively, cDNA can proceed to library preparation and sequencing without genome amplification, as in the case of untargeted high-throughput sequencing.

## 11. Cost-Effectiveness of SARS-CoV-2 Detection in Wastewater

The application of one, or a combination of, molecular-based techniques targeting a region or gene of interest (e.g., RT-PCR), or the whole viral genome (i.e., high-throughput sequencing), will depend on the aims of the study and the resource availability. Table 1 summarizes the advantages and disadvantages of untargeted and targeted or amplicon high-throughput sequencing in comparison with RT-PCR. Establishing novel surveillance activities, as in the case of the wastewater surveillance of SARS-CoV-2, requires trained personnel, as well as reagents and enzymes for molecular assays. This may pose the possibility of deterring already scarce resources for essential surveillance activities, particularly in low-resource settings. Therefore, the cost-benefit ratio of this type of surveillance effort, in relation to other essential activities, should be carefully evaluated before its implementation. When looking at the environmental surveillance of SARS-CoV-2, it may have to be restricted to geographical areas where people are at higher risk of the circulation of the virus, as well as having limited vaccination accessibility [39].

**Table 1.** Advantages and disadvantages of RT-PCR, as well as untargeted and targeted or amplicon-based high-throughput sequencing.

| Method | Advantages | Disadvantages |
|---|---|---|
| RT-PCR | Widely available across laboratories<br>Early detection of low viral titers<br>Quantification of viral load<br>Simultaneous analysis of thousands of samples | False positive results by cross-reactivity<br>False negative results can arise from mutations |
| Untargeted high-throughput sequencing | Viral genetic diversity<br><br>Phylogenetic relationships<br><br>Virus associations with microbiome, virome, and resistome<br>Simultaneous analysis of dozens of samples | Does not provide the same level of genome confidence in wastewater samples compared to clinical samples<br>Higher sequencing depth may be required to determine genetic diversity<br><br>Large computational resources |
| Targeted or amplicon-based high-throughput sequencing | Lower computational power compared to untargeted high-throughput sequencing<br>Viral genetic diversity<br>Viral phylogenetic relationships<br>Simultaneous analysis of dozens of samples | Genomic ends may not be covered |

## 12. Current and Future Applications of SARS-CoV-2 Detection in Wastewater

As described, the SARS-CoV-2 detection in wastewater, using any of the described molecular methods, provides invaluable applications from an epidemiological, biological, and ecological perspective. Some of these current and future applications are discussed below.

### 12.1. Epidemiological Applications

The environmental surveillance of SARS-CoV-2 in wastewater has several epidemiological applications. For instance, the monitoring of SARS-CoV-2 can augment epidemiological information and can complement clinical information when targeting wastewater originating from schools, universities, residential areas, hospitals, prisons, manufacturing and warehouse facilities, airports and airlines, entertainment venues, and gyms, to mention a few. The detection of SARS-CoV-2 in wastewater originating from the mentioned environments can facilitate decision-making and timely public health actions by providing the earliest possible dates for initiating lockdowns and for resuming activities. These types of data also aid in model development and validation to estimate the number of individuals that could be infected within the area or community of interest [59]. This information, in turn, can be important in minimizing the occurrence of a high number of cases, that can limit critical care hospital capacities and long-term care facilities.

The epidemiological applications of monitoring SARS-CoV-2 in wastewater could also benefit from targeting biomarkers that may be significantly elevated in infected in-

dividuals [60]. The rationale behind this is that SARS-CoV-2 infection involves a cascade of immunological and inflammatory mechanisms, and the infection triggers both innate and adaptive immune responses. Various inflammatory cytokines are produced during a SARS-CoV-2 infection. This is also known as a cytokine storm, which evolves through several pathways, leading to the production of interleukin-6 (IL-6) and TNF-alpha [61]. Other parameters that have been explored in SARS-CoV-2 detection include hematological, coagulative, cardiac, and biochemical parameters [60]. Isoprostanes, specifically, have been explored as biomarkers of oxidative stress in wastewater during the SARS-CoV-2 pandemic, showing varying levels, depending on the pandemic timeline [62]. While this area of research has not yet been widely exploited in wastewater monitoring, the characterization of biomarkers resulting from SARS-CoV-2 detection could augment the toolbox of methods for SARS-CoV-2 monitoring in wastewater [60]. Targeting the biomarkers of SARS-CoV-2 infection poses several advantages, including, but not limited to, reduced analytical costs, a broader availability, and an earlier indication of an outbreak, epidemic, or pandemic.

Similar monitoring approaches can be applied to non-wastewater samples, including beach water and freshwater, as well as sand, to determine the potential load of SARS-CoV-2 intact viruses and nucleic acids [63]. The rationale behind monitoring other environments comes from the potential transmission of the virus from infected individuals, both symptomatic and asymptomatic, to environmental sources as the reopening of areas involving water and other recreational activities occur. The epidemiological impact of coming into contact with water used for recreation and consumption remains to be elucidated in more detail; nevertheless, it should not be ignored.

### 12.2. Biological and Ecological Applications

There are several biological and ecological applications for the detection of SARS-CoV-2 nucleic acids in wastewater. For instance, molecular methods (e.g., PCR and high-throughput sequencing) can continue to be optimized, depending on the aims, time, and availability of resources. This may include limitations in detection and quantification, as well as false-positive and false-negative rates, which may be important to understand the prevalence and persistence of SARS-CoV-2 in wastewater across geographical sites. This, in turn, can open the opportunity to determine the persistence, prevalence, and inactivation rate of both RNA and active SARS-CoV-2 particles under different conditions and treatments. This information can also aid in developing a suitable cell culture model to determine the viability and transmission of SARS-CoV-2 in wastewater, as well as other sample types. Finally, the associations and correlations between viral loads, with periods of infectiousness and other biological and environmental surveillance data, can also be understood.

### 13. Conclusions and Future Directions

The present review discussed the current knowledge, as well as several state-of-the-art techniques for the detection of SARS-CoV-2 in wastewater. The detection of SARS-CoV-2 in wastewater could serve as an early indication of infection within specific communities, which could ameliorate having clinical settings diagnose the virus, and could initiate lockdowns in a timely manner. The initial monitoring and surveillance efforts of SARS-CoV-2 in wastewater were supported by the presence of the virus in the feces of both asymptomatic and symptomatic individuals. The presence of the virus in other sample types, such as saliva, has also supported the efforts of monitoring SARS-CoV-2 in wastewater. The current gold standard for the detection of SARS-CoV-2 nucleic acids in wastewater and other sample types is RT-PCR and RT-qPCR. The optimization of the technique has enabled the determination of the prevalence and inactivation rates of the virus in wastewater under different conditions. Sequencing methods, both untargeted and targeted or amplicon-based, have also been proposed and tested, not only to determine the presence of known SARS-CoV-2 variants, but also to determine genetic diversity and phylogenetic relationships. Specific biomarkers of SARS-CoV-2 infection, including cytokines, may also

be included as part of the toolbox of methods for SARS-CoV-2 detection in wastewater and in other environments. Moving forward, both gold standard methods, as well as other techniques that are already part of the toolbox of methods used for the detection of SARS-CoV-2, can be applied for the detection of other viruses affecting humans that can potentially reach wastewater systems in the future.

**Funding:** No funding was received.

**Data Availability Statement:** Not applicable.

**Acknowledgments:** Pablo A. Ortiz for reviewing the draft at early stages.

**Conflicts of Interest:** TS-R is a current employee of Diversigen, a microbiome services company. The present review is a translation ("Deteccion de SARS-CoV-2 en el ambiente: lecciones de aguas residuales") of work that is currently under consideration for publication in a book that will be published in Spanish. This translation was prepared by Tasha M. Santiago-Rodriguez and permission was granted by Pablo A. Ortiz, editor of the book currently under review and that will be published in Spanish.

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
