# Peer review of "The Detection of SARS-CoV-2 in the Environment: Lessons from Wastewater"

_water, doi:10.3390/w14040599_

Round 1
Reviewer 1 Report
The manuscript addresses a current topic in the context of the SARS-CoV-2 pandemic, which aims to identify, by specific methods, in untreated domestic wastewater the RNA of the virus that causes COVID disease.
The information is interesting and provides a well-defined picture of the occurrence of virus in wastewater and the fact that, after treatment, the virus disappears from the treated water.
It would have been necessary to better emphasize the importance of discovering the presence of these viruses in wastewater and additional information on the possible incidence of the disease on contact with them. Also, the references to cases in which RNA was discovered in wastewater are relatively few, and the author, if found only these references in the literature, should hypothesize why there are no more studies.
Author Response
The manuscript addresses a current topic in the context of the SARS-CoV-2 pandemic, which aims to identify, by specific methods, in untreated domestic wastewater the RNA of the virus that causes COVID disease. The information is interesting and provides a well-defined picture of the occurrence of virus in wastewater and the fact that, after treatment, the virus disappears from the treated water.
Thank you to the reviewer for such comments.
It would have been necessary to better emphasize the importance of discovering the presence of these viruses in wastewater and additional information on the possible incidence of the disease on contact with them.
Thank you to the reviewer for this great suggestion. Information regarding the importance of the presence of these viruses in wastewater has now been included (lines 109-112; 124-146)
Also, the references to cases in which RNA was discovered in wastewater are relatively few, and the author, if found only these references in the literature, should hypothesize why there are no more studies.
Thank you to the reviewer for pointing this. Indeed, there are many more studies identifying SARS-CoV-2 nucleic acids in wastewater. Additional references of the detection of the virus nucleic acids in other countries have been added to the new version of the manuscript (lines 160-161).
Reviewer 2 Report
This review speaks about the detection of SARS-CoV-2 in wastewater samples, performed using molecular techniques, with biological and ecological applications, which can be incorporated in future outbreaks, epidemics or pandemics. This study is very interesting, although there are many published works on the same subject. The language is easy and complete. Some suggestion:
- Line 155: The reference 27 talk about “an understanding on the potential role of wastewater in SARS-CoV-2 transmission is largely limited by knowledge gaps in its occurrence, persistence, and removal in wastewater” but not on “the number of individuals that could get infected when in contact with wastewater contaminated with SARS-CoV-2”. The strasmissibily of infection of SARS-CoV-2 through the faecal-oral route does not yet study. Correct and reformulate this sentence.
- Line 145-151: simplify the concept, it is too long and confusing.
- PMID: 33072657, add this study in the introduction section and in reference.
Author Response
This review speaks about the detection of SARS-CoV-2 in wastewater samples, performed using molecular techniques, with biological and ecological applications, which can be incorporated in future outbreaks, epidemics, or pandemics. This study is very interesting, although there are many published works on the same subject. The language is easy and complete.
Thank you to the reviewer for these comments.
Some suggestions:
- Line 155: The reference 27 talk about “an understanding on the potential role of wastewater in SARS-CoV-2 transmission is largely limited by knowledge gaps in its occurrence, persistence, and removal in wastewater” but not on “the number of individuals that could get infected when in contact with wastewater contaminated with SARS-CoV-2”. The strasmissibily of infection of SARS-CoV-2 through the faecal-oral route does not yet study. Correct and reformulate this sentence.
Thank you to the reviewer for pointing out this sentence that seems to be contradicting. The phrase “the number of individuals that could get infected when in contact with wastewater contaminated with SARS-CoV-2” has been removed from the revised version of the manuscript to avoid confusion.
- Line 145-151: simplify the concept, it is too long and confusing.
Lines 145-151 have now been simplified to avoid confusion (new lines 172-182).
- PMID: 33072657, add this study in the introduction section and in reference.
The reference has now been added to the revised version of the manuscript (lines 140-141).